# Indirect ELISA Using Multi–Antigenic Dominants of p30, p54 and p72 Recombinant Proteins to Detect Antibodies against African Swine Fever Virus in Pigs

**DOI:** 10.3390/v14122660

**Published:** 2022-11-28

**Authors:** Dexin Li, Qin Zhang, Yutian Liu, Miaoli Wang, Lei Zhang, Liyuan Han, Xuefei Chu, Guofei Ding, Yingchao Li, Yanmeng Hou, Sidang Liu, Zhiliang Wang, Yihong Xiao

**Affiliations:** 1Department of Fundamental Veterinary Medicine, College of Animal Science and Veterinary Medicine, Shandong Agricultural University, 61 Daizong Street, Tai’an 271018, China; 2Shandong Provincial Key Laboratory of Animal Biotechnology and Disease Control and Prevention, Shandong Agricultural University, Tai’an 271018, China; 3China Animal Health and Epidemiology Center, Qingdao 266032, China; 4Shandong Center Animal Disease Prevention and Control, Jinan 250100, China

**Keywords:** ASFV, antigenic dominant domains, ELISA, p30, p54, p72

## Abstract

African swine fever (ASF) caused by ASF virus (ASFV) is a fatal disease in pigs and results in great economic losses. Due to the lack of available vaccines and treatments, serological diagnosis of ASF plays a key role in the surveillance program, but due to the lack of knowledge and the complexity of the ASFV genome, the candidate target viral proteins are still being researched. False negativity is still a big obstacle during the diagnostic process. In this study, the high antigenic viral proteins p30, p54 and p72 were screened to find the antigenic dominant domains and the tandem His–p30–54–72 was derived. An indirect enzyme–linked immunosorbent assay (iELISA) coated with His–p30–54–72 was developed with a cut–off value of 0.371. A total of 192 clinical samples were detected by His–p30–54–72–coated indirect ELISA (iELISA) and commercial ASFV antibody kits. The results showed that the positive rate of His–p30–54–72–coated iELISA was increased by 4.7% and 14.6% compared with a single viral protein–based commercial ASFV antibody kits. These results provide a platform for future ASFV clinical diagnosis and vaccine immune effect evaluation.

## 1. Introduction

African swine fever (ASF) caused by ASF virus (ASFV) is an acute, hemorrhagic and virulent infectious disease of domestic and wild pigs, characterized by high mortality, often reaching 100%. It was first described in 1921 in Kenya when ASFV was transmitted from wild African boars to domestic pigs [1]. Over the past decade, ASF has spread through the Caucasus, the Russian Federation and eastern Europe and appeared in China in August 2018, resulting in devastating economic losses [2,3]. It was estimated that more than 30 million pigs have been culled from 2018 to 2019, causing estimated economic losses of 2 billion USD for swine production worldwide [4].

The ASFV is an icosahedral, linear double–stranded DNA virus and the only member of the family *Sphaeriidae*, genus *Asfivirus* [5]. The virion is composed of four concentric layers and an external envelope as the fifth layer as it buds through the plasma membrane [4,6,7]. The viral genome varies in length from 170 to 193 kb and contains 150 to 167 open reading frames. The genome consists of a conserved central region of approximately 125 kb and two variable ends containing five multigene families [8]. More than 50 viral proteins induce an antibody response in infected or recovered pigs. The major structural component of the viral capsid p72 as well as the membrane proteins p54, p30, pp62 and p12 are characterized as high antigenic proteins and are usually used as antigens in serological diagnosis, even though these ASFV proteins are insufficient for antibody–mediated protection against different virus strains [9].

Until now, no vaccine has been available against ASFV, so the presence of anti–ASFV antibodies in an animal indicates infection. The serological diagnosis of ASF plays a key role in the surveillance program. Enzyme–linked immunosorbent assay (ELISA), which allows for large–scale testing of samples, is the preferred detection method especially for the detection of pigs that have recovered from subacute and inapparent ASF infection. Several ELISAs coated with p72, p54, p30, pp62 and p12 have been validated for use under different epidemiological situations [10,11,12]. Among them, the p72 protein is mostly used as an antibody detection target for ASFV infection, as it is one of the first identified viral proteins responsible for induction of antibodies post infection and has good immunogenicity with strong conservation [13,14]. The p30 protein is abundantly expressed early in cells and has excellent antigenicity and anti–p30 antibody can be detected at 8 to 12 days post inoculation, which makes it as an important target for early diagnosis of the virus [15]. The anti–p54 antibodies which appears as early as 10 days and persist within the blood of infected animals for several weeks after ASFV infection [16,17,18]. However, the p54 protein has variations in amino acid sequence, which cause false negative results, and therefore, is not usually used as a detection antigen for ASF [11]. These ELISAs may show low diagnostic sensitivity when samples have been poorly preserved or blood samples are used instead of serum [19]. Most of these methods are based on single viral proteins, which may result in a false negative result.

To obtain more accurate detection results of ASF serology, the antigenic dominants of p72, p54 and p30 were screened and tandem expressed in this study. The developed indirect ELISA (iELISA) coated by the tandem expressed showed more sensitivity and specificity compared with a single antigen–coated ELISA, which is useful for ASF surveillance and eradication.

## 2. Materials and Methods

### 2.1. Strains and Plasmids

*Escherichia coli* DH5α competent cells (Takara, Dalian, China) were used for propagation and cloning of plasmids. *E. coli* trans BL21 (DE3) plisse–competent cells and transetta (DE3) chemically competent cells (Transgen biotech, Beijing, China) were used as an expression host. A kanamycin–resistant and polyhistidine (His)–tagged pET–28α (+) and ampicillin–resistant with Glutathione S–transferase (GST) pGEX–6p–1 vectors were used as an expression plasmid in *E. coli*. A kanamycin–resistant with green fluorescent protein (GFP) pEGFP–C1 vector was used for protein expression in mammalian cell lines.

### 2.2. Constructs

The codons of p30, p54 and p72 genes were optimized based on an *E. coli* codon bias and synthesized (Gene Script, Nanjing, China). The truncated p30, p54 and p72 genes were cloned into pET–28a or pGEX–6p–1 vectors, shown in Table 1. The tandem plasmid pET–28a–p30–p54–p72 containing the antigenic dominants was cloned into the pET–28a vector. The eukaryotic expression vector contained p30, p54 and p72 genes with a GFP tag. All the constructs were prepared by molecular cloning technology and confirmed by sequencing.

### 2.3. Expression in E. coli

All the constructs with pET–28a or pGEX–6p–1 vectors were transformed into *E. coli* BL21 competent cells (Transgen biotech, Beijing, China) and cultured at 37 °C in medium containing 50 µg/mL kanamycin or 100 µg/mL ampicillin. When the culture optical density at 600 nm (OD600) reached 0.4 to 0.6, the cells were induced using one mM isopropyl β–D–1–thiogalactopyranoside (IPTG; Solarbio, Beiing, China) for 5 h. The supernatant and inclusion bodies were collected by centrifugation and sonication and the protein expressions were verified by SDS–PAGE.

### 2.4. Purification of Recombinant Proteins

The recombinant His–fused proteins were purified using High Affinity Ni–NTA resin (GenScript USA Inc., Piscataway, NJ, USA) and GST–fused proteins were purified using GSTSep glutathione agarose resin 4FF (Yeasen Biotechnology Co., Ltd., Shanghai, China) according to the manufacturer’s instructions. The insoluble proteins were purified by washing the bound resin column three times with denaturing binding buffer containing 100 mM NaH_2_PO_4_, 10 mM Tris·Cl, 10 mM imidazole and 8 M urea and the bound protein was eluted in elution buffer containing 100 mM NaH_2_PO_4_, 10 mM Tris·Cl, 250 mM imidazole and 8 M urea. The soluble proteins were purified by washing the bound resin column three times with denaturing binding buffer containing 50 mM NaH_2_PO_4_, 50 mM NaCl and 10 mM imidazole and the bound protein was eluted in elution buffer containing 50 mM NaH_2_PO_4_, 300 mM NaCl and 250 mM imidazole.

The recombinant GST–fused proteins were purified when the bound resin column was washed three times with denaturing binding buffer containing PBS consisting of 140 mM NaCl, 2.7 mM KCL, 10 mM Na_2_HPO_4_ and 1.8 mM KH_2_PO_4_ at pH 7.4 and the bound protein was eluted in elution buffer consisting of 50 mM Tris–HCl and 10 mM reduced glutathione at pH 8.0. The purified protein was electrophoresed on 10% to 15% SDS–PAGE and visualized by Coomassie brilliant blue G250 staining.

### 2.5. Preparation of Mouse and Rabbit Polyclonal Antibody

The purified p30, p54 and p72 proteins were emulsified with Freund’s adjuvant to immunize mice and rabbits. All the purified proteins were administrated into six–week–old female BALB/C mice with a dose of 100 µg protein per mice or six–week–old female New Zealand white rabbits with a dose of 1 mg protein per rabbit. Complete Freund’s adjuvant was used for the first immunization and incomplete Freund’s adjuvant was used for the booster injection two weeks later. Two weeks after the third immunization, blood was collected from the tail vein of mice to detect antibody levels.

### 2.6. Western Blot Analysis

The products with or without IPTG induction and the purified proteins were boiled with 5 × loading buffer for 10 min and then separated by SDS–PAGE and transferred to polyvinylidene fluoride membrane (Millipore, Darmstadt, Germany). After blocking with 5% BSA for 2 h, the target proteins were probed using anti–His monoclonal antibody diluted 1:2000 in 5% BSA (CWbio Co., Ltd., Beijing, China), anti–GST monoclonal antibody diluted 1:1000 in 5% BSA (MerryBio, Nanjing, China) and anti–Mouse or anti–Rabbit p30, p54 and p72 polyclonal antibodies diluted 1:300 in 5% BSA for 2 h. The membrane was then incubated with HRP–conjugated secondary antibody (Beyotime Institute of Biotechnology, Haimen, China) for 1 h. Protein bands were visualized using the Clarity Western ECL substrate (Bio–Rad) by NcmECL Ultra (NCM Biotech Co. Ltd., Suzhou, China).

### 2.7. Screening of Dominant Epitopes

According to the antigenic index of the p30, p54 and p72 proteins, three dominant epitopes in p30, i.e., 1 to 66 aa, 78 to 132 aa and 134 to 174 aa; 54 to 124 aa in p54; and four dominant epitopes in p72, i.e., 12 to 89 aa, 139 to 324 aa, 445 to 524 aa and 552 to 647 aa, were predicted in Figure 1 (Jameson and Wolf, 1988). All the corresponding genes were cloned into prokaryotic expression vectors and expressed in *E. coli*, as listed in Table 1. The dominant epitopes were screened using prepared mouse or rabbit anti–p30, p54 and p72 and clinical ASFV positive serum by Western blot.

### 2.8. Expression of the Tandem Antigenic Dominant Protein His–p30–54–72

The gene fragments corresponding to the dominants p30 78–132 aa, p30 134–174 aa, p54 54–124 aa, p72 12–89 aa, p72 139–324 aa and p72 445–524 aa were linked in sequence and inserted into the pET–28a vector in Table 1. To ensure the feasibility and structure of each dominant, a linker containing 10aa was added between each. After confirming with gene sequencing, the correct recombinant plasmids were transformed into *E. coli* BL21–competent cells and cultured at 37 °C in LB medium containing 50 µg/mL kanamycin. When the culture OD600 reached 0.4 to 0.6, the cells were induced using 1 mM IPTG. The supernatant and inclusion bodies were collected and the protein expressions were verified by 12% SDS–PAGE. The tandem antigenic dominant protein His–p30–54–72 was purified using High Affinity Ni–NTA resin according to the manufacturer’s instructions. The purification protein was electrophoresed on SDS–PAGE and visualized by Coomassie brilliant blue G250 staining.

### 2.9. Optimization of the p30–54–72 Coated iELISA

To develop the p30–54–72 coated iELISA, each component including the coating buffer, blocking agent and determination of the antigen concentration as 25, 50, 100, 200, 300, 400, 500, 800 and 1200 ng/well and test sera dilutions were determined separately. The optimal antigen concentration and the serum dilution for the iELISA were determined by standard checkerboard titration of serially diluted fusion proteins. Under the optimized conditions of ELISA, the cut–off values were obtained. The cut–off values were determined by testing 82 positive and 38 negative serum samples, which were confirmed by commercial ELISA. After being analyzed by TG–ROC software [20].

### 2.10. Serum Samples

A set of 82 positive and 38 negative serum samples and a total of 192 clinical serum samples collected from pig herds with suspected ASFV infection were detected and preserved by the National African Swine Fever Reference Laboratory of the China Animal Health and Epidemiology Center using commercial ASFV antibody ELISA kits (Beijing, Jinnuo Baitai Biotechnology Co., Ltd. (Beijing, China) (JN60915) (based on p30 protein); Qingdao RealVet Bio–Technology Co., Ltd. (Qingdao, China) (ASF.K001/5); Tianjing Biovetest Biotechnology Co., Ltd. (Tianjing, China) (BVE0201, BVE0202) (based on CD2v and p30)). The antigen was identified as ASFV genotype II, lately. The positive and negative serum samples were used for the detecting of the cut–off value of the p30–54–72 coated iELISA. The 192 clinical samples were detected by commercial ASFV antibody ELISA kits (Beijing Jinnuo Baitai Biotechnology Co., Ltd. (JN60915) (based on p30 protein); Qingdao RealVet Bio–Technology Co., Ltd. (ASF.K001/5)) (based on p72 protein)). The detection results are listed in Table 2.

### 2.11. Application of the p30–54–72 Coated iELISA

Ninety–six–well reaction plates were coated with 500 ng/well of purified protein overnight at 4 °C. The plates were then washed four times with PBST consisting of 135 mM NaCl, 1.3 mM KCL, 3.2 mM Na_2_HPO_4_ and 0.5 mM KH_2_PO_4_, 0.05% (*v/v*) Tween–20, blocked in 2.5% skim milk in PBST for 1 h at 37 °C and rinsed for four times in PBST. The dilution of the swine sera was 1:40 and 100 µL of diluted serum to be evaluated was added and incubated for 1 h at 37 °C. After washing the plate four times with PBST, samples were then plated with HRP–conjugated goat anti–pig IgG diluted 1:2000 in PBS and incubated for 1 h at 37 °C, followed by washing with PBST for four times. Tetramethylbenzidine (TMB) buffer was added and incubated for 10 to 30 min at room temperature. Lastly, 50 µL 3M H_2_SO_4_ was added to each well and the concentration was determined at an optical density (OD) of 450 nm.

## 3. Results

### 3.1. Expression and Purification of His–Fused p30, p54 and p72 Proteins

The codons of p30, p54 and p72 genes were optimized based on an *E. coli* codon bias and synthesized. The optimized genes were separately cloned into pET–28a vector and expressed in *E. coli*. and purified. The Recombinant His–fused p30, p54 and p72 proteins were all successfully expressed in *E. coli* BL21 cells with predicted molecular weight of 22.4, 19.8 and 73.2 KDa, respectively. Further analysis showed His–p30, His–p72 presented in the inclusion bodies and His–p54 presented in the supernatant, as shown in Figure 2A. The recombinant proteins were then purified by Ni–IDA resin and identified by Western blot using anti–His mAb, as shown in Figure 2B.

### 3.2. Preparation of the Anti–p30, p54 and p72 Polyclonal Antibodies

Five female BALB/c mice and three New Zealand rabbits were separately immunized with purified p30, p54 and p72 proteins to generate polyclonal antibodies. All the immunized animals produced specific antibodies with tires of 1:10^4^ to 1:10^5^, as shown in Figure 3A. To confirm the specificity of the antibodies, eukaryotic expression vectors containing p30, p54 and p72 genes with GFP tags were transfected into 293T cells. Western blot results showed that the polyclonal antibodies produced in mice and rabbits could recognize the corresponding antigens with molecular weights of 56, 50 and 98 KDa, respectively, shown in Figure 3B,C. All these results indicated that the p30, p54 and p72 antibodies were prepared successfully.

### 3.3. Preparation of Truncated p30, p54 and p72 Proteins

According to the antigenic index of p30, p54 and p72 proteins, three fragments of p30, one fragment of p54 and four fragments of p72 were subcloned into pET–28a or pGEX–6p–1 vectors, as shown in Figure 1. To generate successful expression, p30 1–66 aa, p30 78–132 aa, p30 134–174 aa, p72 12–89 aa and p72 445–524 aa were cloned into the pGEX–6p–1 vector fused with GST and the predicted molecular weights were 33.41, 32.35, 30.82, 34.75 and 35.34 KDa, respectively. The p54 54–124 aa, p72 139–324 aa and p72 552–647 aa were subcloned to the pET–28a vector with a His tag with predicted molecular weights of 10.4, 21.21 and 11.54 KDa, respectively. The expression of truncated proteins was confirmed by SDS–PAGE and Western blot, as shown in Figure 4A,B. The results showed that all the truncated proteins in Table 1 were expressed in the insoluble form.

### 3.4. Screening the Dominant Antigenic Domains

The truncated proteins were analyzed by mice and rabbit polyclonal antibodies together with the positive serum from pigs. The results showed that p30 78–132 aa, p30 134–174 aa, p54 54–124 aa, p72 12–89 aa and p72 445–524 aa could be recognized by swine serum strongly, as shown in Figure 5A. All the fragments except p72 552–647 aa could be recognized by mouse and rabbit polyclonal antibodies. Based on these results, all the fragments except for fragments of p30 1–66 aa and p72 552–647 aa could induce the production of the antibodies in distinct species, as shown in Figure 5B,C. These results showed that p30 78–132 aa, p30 134–174 aa, p54 54–124 aa, p72 12–89 aa, p72 139–324 aa and p72 445–524 aa could evoke antibody production.

### 3.5. The Tandem Expression of the Dominant Antigenic Domains of p30, p54 and p72 Proteins

The gene fragments corresponding to p30 78–132 aa, p30 134–174 aa, p54 54–124 aa, p72 12–89 aa, p72 139–324 aa and p72 445–524 aa were linked in sequence and inserted into the pET–28a vector. A linker containing 15 aa was added between each fragment. After confirming with gene sequencing, the correct recombinant plasmid was expressed in *E. coli*. The results showed that a target band with predicted molecular weight of 62 KDa named His–p30–54–72 presented in the inclusion bodies, as shown in Figure 6A. The recombinant proteins were then purified by Ni–IDA resin and identified by Western blot using anti–His mAb, as shown in Figure 6B. To further confirm the antigenic activity, the purified His–p30–54–72 was detected by the prepared rabbit anti–p30, p54 and p72 polyclonal antibodies. Western blot results in Figure 6C showed that His–p30–54–72 could be recognized by the rabbit polyclonal antibodies specifically, showing that the p30–p54–p72 was expressed successfully.

### 3.6. Optimization of the His–p30–54–72 Coated ELISA

After detection by draught board titration tests, the optimal concentrations of the coating antigen were set as 500 ng/well and the dilution of the swine sera was 1:40 for the p30-p54-p72-coated ELISA. The cut-off values were determined by testing 82 positive and 38 negative serum samples which were confirmed by commercial ELISA. After being analyzed by TG-ROC software [20], a cut-off value of 0.371 at OD450 nm was calculated, with a sensitivity value of 97.5% and specificity value of 95.4%. The intra-assay and inter-assay variabilities were evaluated by comparing the OD ratios of quadruplicate serum samples that were evaluated repeatedly between and within ELISA plates. By evaluating five selected serum samples, the inter-assay coefficients of variability (CVs) were observed, ranging from 0.68% to 7.70%, with a median value of 3.7%, and the intra-assay CV was observed, ranging from 4.3% to 8.2%, with a median value of 4.6%. These data indicated that the His-p30-54-72-coated ELISA was repeatable with low variations.

### 3.7. Application of His–p30–54–72–Coated ELISA

To validate the His–p30–54–72–coated ELISA, 192 serum samples were detected, which were confirmed by commercial ELISA kits. The results showed that among the 192 clinical serum samples, 149 samples were detected by the His–p30–54–72–coated ELISA, with a positive rate of 77.6% shown in Table 2. In total, 140 samples were detected by the p30–based commercial iELISA, with a positive rate of 72.9%, and 121 samples were detected by the p72–based commercial ELISA, with a positive rate of 63.0%. Compared with commercial kits, an additional 9 and 28 more samples, respectively, were detected by the His–p30–54–72–coated iELISA and the positive rates were increased by 4.7% and 14.6%, respectively. Of the positive samples detected by the His–p30–54–72–coated iELISA, 121 samples were positive for both the His–p30–54–72–coated and p72–based commercial ELISAs. The positive coincidence rate of the His–p30–54–72–coated iELISA and p72–based commercial ELISA was 100%. Among the 71 negative samples detected by the p72–based commercial ELISA, 28 samples were detected as positive by the His–p30–54–72–coated iELISA. The negative coincidence rate of the His–p30–54–72–coated iELISA and p72–based commercial ELISA was 60.6%. Of the positive samples detected by the His–p30–54–72–based iELISA, 137 samples were positive for both the His–p30–54–72–coated and p30–based commercial iELISAs. The positive coincidence rate of the His–p30–54–72–based iELISA and p30–based commercial ELISA was 97.9%. Among the 52 negative samples detected by the p30–based commercial iELISA, 12 samples were detected as positive by the His–p30–54–72–coated iELISA. The negative coincidence rate of the His–p30–54–72–based iELISA and p30–based commercial ELISA was 76.9%.

## 4. Discussion

The ASFV is quite complexed and not well understood [21,22]. On 3 June 2022, Vietnam officially announced the release of the world’s first ASFV vaccine, but on August 24, the vaccine was abandoned due to more than a dozen swine deaths after vaccination. Currently, a commercial vaccine is still not available and this indicates that the control of ASF was a challenging task. The surveillance program may last for a long time, so an ELISA test with high sensitivity and specificity is urgently needed.

Serological diagnosis is indispensable in the program of ASF surveillance. Since its appearance in the 1970s, ELISA has been continually improved and widely used in the clinical detection [23]. It is a powerful tool in detecting the presence of antigen–specific antibodies to improve handling and response times. The key component of an ELISA is viral proteins with high antigenic character. The structure of the ASFV proteins p30, p54 and p72 was reported to be highly antigenic and therefore are usually used as antigens in serological diagnosis. Most commercial ELISA kits are based on single viral proteins, which may result in a false negative result [10,24,25]. In this study, an iELISA coated with the tandem proteins containing the main antigenic dominants of p30, p54 and p72 was developed. The antibody in pigs induced by ASFV infection is time–dependent and antibodies specific to certain antigenic domains may be omitted. Mice and Rabbits polyclonal antibodies against p30, p54 and p72 proteins were prepared and used for screening the main antigenic dominants. Two domains in p30, one domain in p54 and three domains in p72 were confirmed as the antigenic dominant domains. All the gene fragments of the antigenic dominant domains were linked together to a tandem protein His–p30–54–72 with a linker containing 15 aa between each domain. Compared with the ELISA with a single viral protein, the sensitivity was improved greatly, which indicates the His–p30–54–72–coated iELISA’s suitability for the serological diagnosis of ASFV.

Compared with commercial kits, the His–p30–54–72–coated iELISA can detect more positive samples as shown in Table 2. All the positive samples were detected negatively by commercial antibody kits of porcine reproductive and respiratory syndrome virus, swine fever virus and porcine pseudorabies virus (Data not shown), indicating the good specify of the His–p30–54–72–coated iELISA. However, unavailable serum samples from experimental trial in pigs were not detected, and further validation was needed to improve this method. A p54–coated iELISA developed in this lab was also used to detect clinical samples (data not shown). Of the 192 samples, a total of 137 (71.4%) positive samples were detected and a positive coincidence rate was detected with the His–p30–54–72–coated samples.

iELISA accuracy was 98.5%. Of the 55 negative samples detected by the p54–coated iELISA, 14 samples were detected as positive by the His–p30–54–72–coated iELISA. Comparing the His–p30–54–72–coated iELISA with the p54–coated iELISA, the number of positive samples detected increased by 12 and the positive detection rate increased by 6.3%.

## 5. Conclusions

An iELISA coated with the tandem His–p30–54–72 protein containing the main antigenic dominants of p30, p54 and p72 of ASFV was developed. The His–p30–54–72–coated iELISA was more sensitive with high specificity than those of single–viral protein–coated ELISAs and provides a powerful platform for future ASFV clinical diagnosis and vaccine immune effect evaluation. Combined with real–time PCR, it will be a powerful tool in the ASF surveillance program.

## Figures and Tables

**Figure 1 viruses-14-02660-f001:**
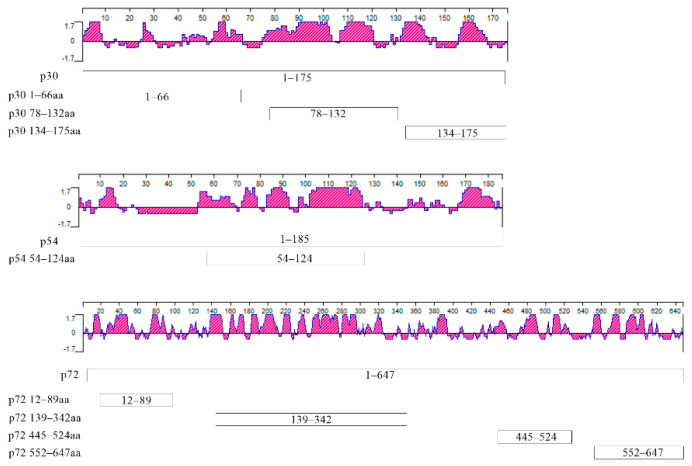
Jameson–Wolf protein antigenicity profile for p30, p54 and p72. The X–axis values are amino acid residues for each protein; the Y–axis represents the antigenic index (Jameson and Wolf, 1988). Positions of each truncated recombinant proteins are shown in boxes. p30 1–66aa, 78–132aa and 134–174aa contain amino acids of p30 from 1 to 66, 78 to 132 and 134 to 174, respectively. p54 54–124aa contains amino acids of p54 from 54 to 124. p72 12–89aa, p72 139–324aa, p72 445–524aa and p72 552–647aa contain amino acids of p72 from 12 to 89, 139 to 324, 445 to 524 and 552 to 647, respectively.

**Figure 2 viruses-14-02660-f002:**
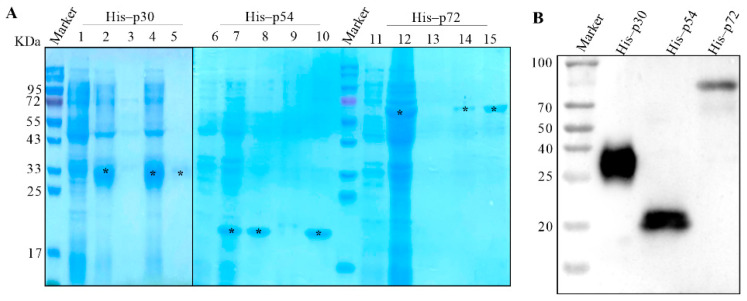
Identification of His–fused p30, p54 and p72 proteins. (**A**) Identification of purification of His–fused p30, p54 and p72 proteins by SDS–PAGE. All the samples were electrophoresed on SDS–PAGE and visualized by coomassie brilliant blue G250 staining. Here, 1, 6, 11 represent whole cell lysates without IPTG induction, 2, 7, 12 represent whole cell lysates with IPTG induction, 3, 8, 13 represent the expression of His–fused protein from the supernatant, 4, 9, 14 represent the expression of His–fused protein from the inclusion bodies and 5, 10, 15 represent purified proteins. (**B**) Western blot identification of the His–fused p30, p54 and p72 proteins. The purified His–fused p30, p54 and p72 proteins were separated by SDS–PAGE and transferred to PVDF membrane. The target proteins were probed with anti–His antibody. *: the target bands.

**Figure 3 viruses-14-02660-f003:**
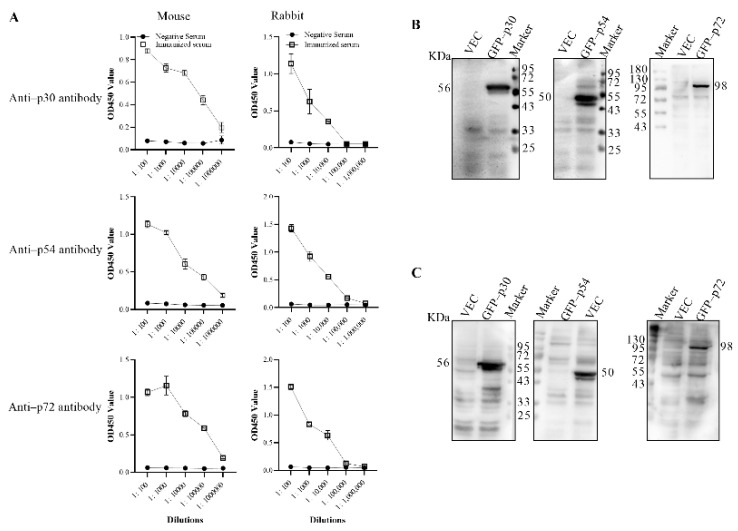
Confirmation of mouse and rabbit polyclonal antibodies. (**A**) The titer of the polyclonal antibodies produced in mouse and rabbit. X–axis was the sera with different dilution ratios. Y–axis on the left was OD values generated by p30, p54, p72–coated iELISA for the detection of antibody responses against p30, p54, p72, respectively. Each point on each dilution represents the mean OD values obtained from 3 mice and 3 rabbit serum samples. Results were performed as the means ± SD and analyzed with Graph–PadPrism8 software (San Diego, CA, USA). Western blot was used to confirm the specificity of the mouse (**B**) or rabbit (**C**) polyclonal antibodies. Eukaryotic expression vector containing p30, p54 and p72 genes with GFP tag or pEGFP–C1 vector were transfected into 293T cells. The cell lysates were immunoblotted with mouse or rabbit polyclonal anti–p30, p54 or p72 antibody, respectively.

**Figure 4 viruses-14-02660-f004:**
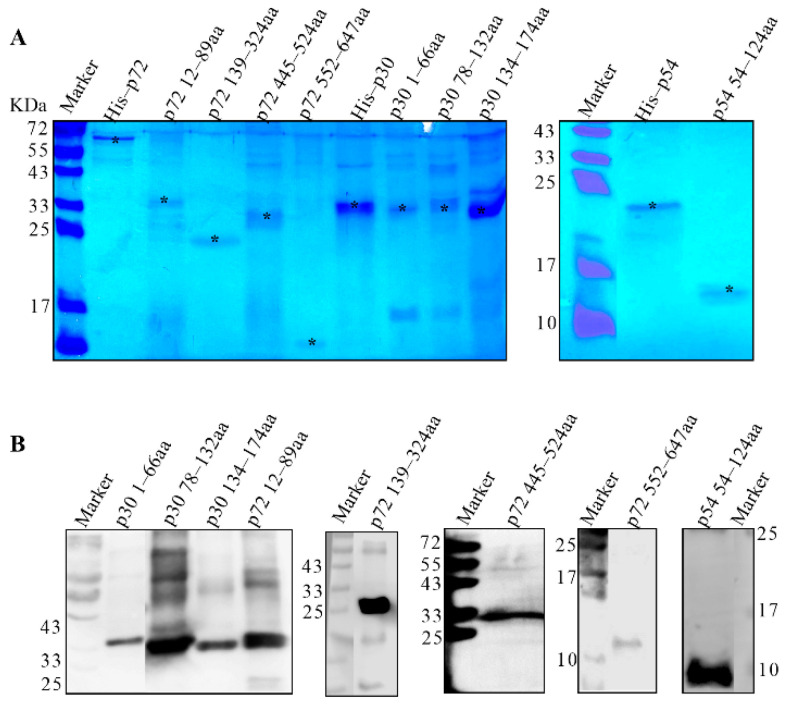
Identification of truncated proteins. (**A**) Identification the expression of truncated proteins by SDS–PAGE. The whole cellular proteins with induction were electrophoresed on 10–15% SDS–PAGE and visualized by coomassie brilliant blue G250 staining. The predicted molecular weights with a GST tag of p30 1–66aa, p30 78–132aa, p30 134–174aa, p72 12–89aa and p72 445–524aa were 33.41, 32.35, 30.82, 34.75 and 35.34 KDa, respectively. The predicted molecular weights with an His tag of p54 54–124aa, p72 139–324aa and p72 552–647aa were 10.4, 21.21 and 11.54 KDa, respectively. (**B**) Identification the expression of truncated proteins by Western blot. The purified proteins of p30 1–66aa, p30 78–132aa, p30 134–174aa, p72 12–89aa and p72 445–524aa were separated and probed with anti–GST monoclonal antibody. The purified proteins p54 54–124aa, p72 139–324aa and p72 552–647aa were separated and probed with anti–His monoclonal antibody. *: the target bands.

**Figure 5 viruses-14-02660-f005:**
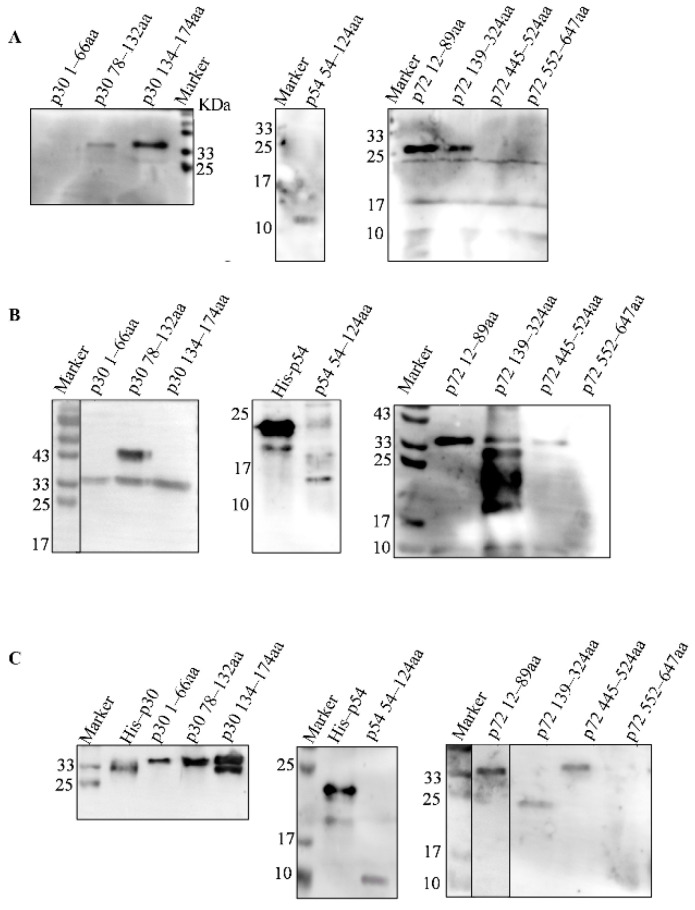
Screening the antigenic dominants of p30, p54 and p72 proteins. Screening the dominant antigenic domains. All the truncated proteins were separated by SDS–PAGE and transferred to PVDF membrane. The target proteins were probed with ASFV positive pig serum (**A**), mouse anti–p30, –p54 and –p72 polyclonal antibody (**B**) and rabbit anti–p30, –p54 and –p72 polyclonal antibody (**C**).

**Figure 6 viruses-14-02660-f006:**
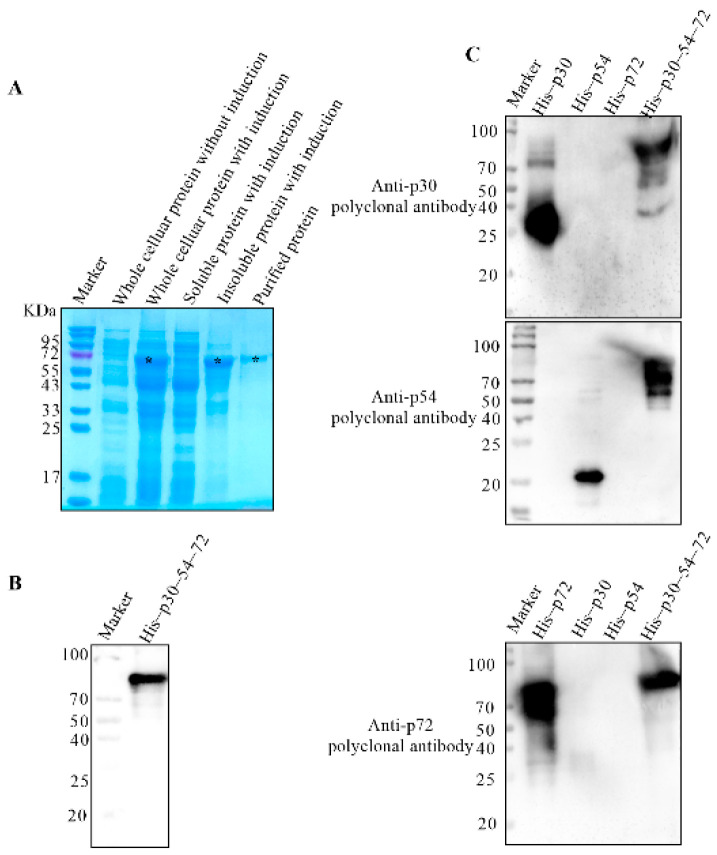
Identification of His–p30–p54–p72 protein. The cellular or purified proteins were electrophoresed on 10% SDS–PAGE and visualized by coomassie brilliant blue G250 staining (**A**) or were transferred to PVDF membrane probed with anti–His monoclonal antibody (**B**). (**C**) Identification of the His–p30–p54–p72 protein by rabbit anti–p30, –p54 and –p72 polyclonal antibody. *: the target bands.

**Table 1 viruses-14-02660-t001:** Primers used for constructs.

Constructs	Primer (5′–3′)	Vector	Insoluble/Soluble
pET–28a	F: CAAGCTTGCGGCCGCACT		
	R: TCGACGGAGCTCGAATTCG		
EGFP–C1	F: GTCGACGGTACCGCGGGC		
	R: TGCAGAATTCGAAGCTTGAGC		
pGEX–6P–1	F: GTCGACTCGAGCGGCCGCA		
	R: CCCGGGAATTCCGGGGATC		
pEGFP–C1–p30	F: CTCAAGCTTCGAATTCTGCAATGGATTTTATTTTAAATATATCCAT	pEGFP–C1	Eukaryotic expression
	R: GGGCCCGCGGTACCGTCGACTTATTTTTTTTTTAAAAGTTTAA		
pEGFP–C1–p54	F: CTCAAGCTTCGAATTCTGCAATGGATTCTGAATTTTTTCAA	pEGFP–C1	Eukaryotic expression
	R: GGGCCCGCGGTACCGTCGACTTACAAGGAGTTTTCTAGGTCT		
pEGFP–C1–p72	F: CTCAAGCTTCGAATTCTGCAATGGCATCAGGAGGAGCTT	pEGFP–C1	Eukaryotic expression
	R: GGGCCCGCGGTACCGTCGACTTAGGTACTGTAACGCAGCACA		
pGEX–6P–1–p30 (1–66aa)	F: GATCCCCGGAATTCCCGGGATGGACCTGCGTAGCAGCAG	pGEX–6P–1	Insoluble
	R: TGCGGCCGCTCGAGTCGACTTCCTCTTGCGCCTGGTGC		
pGEX–6P–1–p30 (78–132aa)	F: GATCCCCGGAATTCCCGGGGAGACCGAAAGCAGCGCGAG	pGEX–6P–1	Insoluble
	R: TGCGGCCGCTCGAGTCGACGATGTGTTGAACGGTTTTCTGC		
pGEX–6P–1–p30 (134–174aa)	F: GATCCCCGGAATTCCCGGGCAGTATGGCAAGGCGCCGG	pGEX–6P–1	Insoluble
	R: TGCGGCCGCTCGAGTCGACCAGTTTGATAACCATCA		
pET–28a–p54 (54–124aa)	F: CCGAATTCGAGCTCCGTCGACGTCCGGCGACCAACCGTC	pET–28a	Soluble
	R: CGAGTGCGGCCGCAAGCTTGATAGGTGTTACGTTGACGC		
pGEX–6P–1–p72 (12–89aa)	F: GATCCCCGGAATTCCCGGGGATGGGAAGGCCGACAAG	pGEX–6P–1	Insoluble
	R: TGCGGCCGCTCGAGTCGACAAGCTTGTTTCCCAAGGTG		
pET–28a–p72 (139–324aa)	F: CCGAATTCGAGCTCCGTCGACGCAACGGATATGACTGGG	pET–28a	Insoluble
	R: CGAGTGCGGCCGCAAGCTTGCTGATAGTATTTAGGGGTTTG		
pGEX–6P–1–p72 (445–524aa)	F: GATCCCCGGAATTCCCGGGCACACCAACAATAACCACCAC	pGEX–6P–1	Insoluble
	R: TGCGGCCGCTCGAGTCGACATCCGATATAGATGAACATG		
pET–28a–p72 (552–647aa)	F: CCGAATTCGAGCTCCGTCGAAAGTTTCCATCAAAGTTCTG	pET–28a	Insoluble
	R: CGAGTGCGGCCGCAAGCTGAAGTTTCCATCAAAGTTC TG		
Linker–p30 (78–132aa)	F: CCGAATTCGAGCTCCGTCGAGAGACCGAAAGCAGCGCG		
	R: GATCCTCCACCTCCTGATCCACCTCCACCGATGTGTTGAACGGTTT		
Linker–p30 (134–174aa)	F: GATCAGGAGGTGGAGGATCACAGTATGGCAAGGCGCCG		
	R: AGAACCACCGCCACCCGAGCCGCCACCGCCCAGTTTGATAACCA		
Linker–p54 (54–124aa)	F: GGCGGTGGCGGCTCGGGTGGCGGTGGTTCTCGTCCGGCGACCAA		
	R: CGATCCGCCTCCACCGGAACCTCCGCCTCCATAGGTGTTACGTTGA		
Linker–p72 (12–89aa)	F: GGAGGCGGAGGTTCCGGTGGAGGCGGATCGGATGGGAAGGCCG		
	R: GGAGCCTCCGCCGCCAGATCCGCCTCCCCCAAGCTTGTTTCCCAA		
Linker––p72 (139––324aa)	F: CTGGCGGCGGAGGCTCCCGCAACGGATATGACTGGGA		
	R: AGAACCACCGCCACCCGAGCCGCCACCGCCCTGATAGTATTTAG		
Linker––p72 (445––524aa)	F: CGGGTGGCGGTGGTTCTCACACCAACAATAACCACCAC		
	R: CGAGTGCGGCCGCAAGCTTGATCCGATATAGATGAACATGCGT		

**Table 2 viruses-14-02660-t002:** Clinical samples from detected by different ELISAs.

Target Proteins	a/b/p30–54–72 (+)	a/b (+)/p30–54–72(+)	a/b (−)/p30–54–72 (−)	a or b (−)/p30–54–72(+)
p30 ^a^	140 (72.9%)	137 (97.9%)	40 (76.9%)	9 (4.7%)
p72 ^b^	121 (63%)	121 (100%)	43 (60.6%)	28 (14.6%)
His–p30–p54–p72	149 (77.6%)	/	/	/

Note: ^a^ Detected by ELISA kit from Beijing Jinnuo Baitai Biotechnology Co., Ltd. (JN60915) based on p30 protein; ^b^ Detected by ELISA kit from Qingdao RealVet Bio–Technology Co., Ltd. (ASF.K001/5) based on p72 protein; +: positive; −: negative.

## Data Availability

The data that support the findings of this study are available from the corresponding author upon reasonable request.

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
