# Peer review of "Indirect ELISA Using Multi–Antigenic Dominants of p30, p54 and p72 Recombinant Proteins to Detect Antibodies against African Swine Fever Virus in Pigs"

_viruses, 2022, doi:10.3390/v14122660_

Round 1
Reviewer 1 Report
Comments for the manuscript titled “Indirect ELISA using multi-antigenic dominants of p30, p54 and p72 recombinant proteins to detect antibodies against African swine fever virus in pigs”
In this manuscript, an iELISA coated with His-p30-54-72 was developed. Overall, this manuscript is well written. The following changes should be introduced in the manuscript before its acceptance.
1. There is too much overlap between the description of result and discussion. Please modify it.
2. English should be improved.
Reviewer 2 Report
I reviewed the manuscript entitled “Indirect ELISA Using Multi-Antigenic Dominants of p30, p54 and p72 Recombinant Proteins to Detect Antibodies Against African Swine Fever Virus in Pigs”. In this study, authors present the development and validation an ELISA using a HIS-P30-p54-p72 antigen for the detection of antibodies of ASFV. The results of this research showed the superiority of this ELISA in the detection of positive samples when compared with commercial ELISAS.
Overall, I think the design proposed by the authors may be a valuable tool for the diagnosis of ASFV. However, I consider that different issues should be properly addressed before considering this study for publication.
A) The introduction should be improved, including more specific information regarding previous results using the antigens proposed in this study for the detection of antibodies against ASFV. I suggest authors considering this reference: Detection of African Swine Fever Virus Antibodies in Serum and Oral Fluid Specimens Using a Recombinant Protein 30 (p30) Dual Matrix Indirect ELISA. PLoS One. 2016 Sep 9;11(9):e0161230. doi: 10.1371/journal.pone.0161230. PMID: 27611939; PMCID: PMC5017782.
B) In the methods, include more information regarding the serum samples (positive, Negative and clinical samples) included for the validation of this study. Do positive and negative samples come from experimental trails? Same for clinical samples. Do these samples come from diagnosed animals? Also, please, indicate if the genetic origin of the virus o viruses associated with all these samples is known?
C) In the results section, the validation using the positive and negative samples is not clear. Please, show a clear information regarding the comparison with the commercial ELISAs. Is Figure 6 associated with the specificity and sensibility as stated in the text? I didn’t find table 2. Please, include the complete information.
D) Improve the discussion including information about the limitation of the validation presented in this study. It is necessary the inclusion of serum samples positive to other porcine viruses to assess potential cross reactions. Also, include samples from animals at different stages of infection. The use of positive samples coming from animals infected with viruses form different genetic origin is needed for a complete validation. Please discuss all these aspects.
Round 2
Reviewer 2 Report
I like to thank the authors for their responses. For me is not still clear the answer to my question about the origin of the positive and the negative samples. Are they experimental samples? Please specify. Also, table 2 is not clear for me. I suggest the author to present two tables. One using the set of positive and negative samples and comparing the commercial ELISA with the ELISA developed in this study. The second table presenting the same comparison using the clinical samples. I think the information in the first table is a critical component for the validation.
Round 3
Reviewer 2 Report
Thank you to the authors for their responses. I think the fact that serum samples classified as positive or negative to perform the validation of this technique did come from an experimental trial in pigs should be clarified. If these samples came form clinical samples, it should be mentioned as a limitation in the discussion, mentioning that a further validation is needed to improve this method. In this context, I am especially concerned about negative samples. Is it possible to include in the validation a set of serum samples coming from a reference source? Also, in this context, it is important to include in the results the methodology used to classify as positive and negative the clinical samples used for the standardization of these method. All of them were positive by ELISA tests for the detection of P72 and P30 and P54?
